# Humic Substances from Different Sources Modulate Salicylic Acid-Mediated Defense in Plants Infected by Powdery Mildew

**DOI:** 10.3390/plants14243854

**Published:** 2025-12-17

**Authors:** Rakiely M. Silva, Vicente Mussi-Dias, Fábio L. Olivares, Lázaro E. P. Peres, Luciano P. Canellas

**Affiliations:** 1Núcleo de Desenvolvimento de Insumos Biológicos para Agricultura (NUDIBA), Universidade Estadual do Norte Fluminense Darcy Ribeiro (UENF), Ave Alberto Lamego 2000, Campos dos Goytacazes 28013-602, Brazil; fabioliv@uenf.br (F.L.O.); canellas@uenf.br (L.P.C.); 2Laboratório de Entomologia e Fitopatologia, Centro de Ciências e Tecnologias Agropecuárias, Universidade Estadual do Norte Fluminense Darcy Ribeiro, Campos dos Goytacazes 28013-602, Brazil; vicmussi@uenf.br; 3Departamento de Ciências Biológicas, Escola Superior de Agricultura ‘‘Luiz de Queiroz’’ (ESALQ), Universidade de São Paulo (USP), Piracicaba 05508-090, Brazil; lazaro.peres@usp.br

**Keywords:** induced resistance, hormone, biostimulants

## Abstract

Modern agriculture relies heavily on chemical inputs to sustain productivity, yet their intensive use poses environmental and health risks. Sustainable strategies based on biostimulants have emerged as promising alternatives to reduce agrochemical dependence. Among these compounds, humic substances (HS) stand out for their ability to modulate plant growth and activate defense responses. This study aimed to evaluate the effects of HS from different sources—vermicompost (Vc) and peat (Pt)—on the salicylic acid (SA)-mediated defense pathway in tomato plants (*Solanum lycopersicum* cv. Micro-Tom) infected with *Oidium* sp. The HS were characterized by solid-state ^13^C CPMAS NMR to determine the relative distribution of carbon functional groups and structural domains, including alkyl, O-alkyl, aromatic, and carbonyl carbon fractions, as well as hydrophobicity-related indices. Enzymatic activities of lipoxygenase, peroxidase, phenylalanine ammonia lyase, and beta 1,3-glucanase were determined spectrophotometrically, and RT-qPCR quantified gene transcription levels involved in SA signaling and defense (*MED25*, *MED16*, *MED14*, *NPR1*, *ICS*, *PAL*, *LOX1.1*, *MYC2*, *JAZ*, *jar1*, *CAT*, *POX*, *SOD*, *APX*, *ERF*, *PR-1*, *PR-2*, *PR-4 e PR-5*). Both HS significantly reduced disease severity and activated key SA-related defense genes, including the regulatory gene *NPR1* and the effector genes *PR1*, *PR2* and *PR5*, with Pt providing greater protection. Notably, HS amplified defense-related gene expression and enzymatic activities specifically under infection, showing a stronger induction than in non-infected plants. These results demonstrate that structural differences among HS drive distinct and enhanced defense responses under pathogen challenge, highlighting their potential as sustainable tools for improving plant immunity in agricultural systems.

## 1. Introduction

Plant health is essential for maintaining global food security. Climate change has intensified abiotic and biotic stresses in agricultural systems, increasing the vulnerability of crops to pathogens and reducing global food security [1,2]. Tomato (*Solanum lycopersicum*) is one of the most economically important horticultural crops worldwide, with high demand for both fresh consumption and processed products [3]. However, tomato production is particularly sensitive to fungal diseases, which are major contributors to yield instability. Among these pathogens, *Oidium* spp., the causal agent of powdery mildew, is responsible for significant economic losses due to reduced photosynthetic capacity, premature leaf senescence, and decreased fruit quality [4]. Tomato has also been widely used as a model system to investigate defense-response mechanisms and hormone-mediated immunity [5], further supporting its relevance for studies on biotic stress responses. In modern agriculture, disease management still primarily relies on direct pathogen control, predominantly via pesticide application. Although effective in the short term, such practices have caused significant environmental impacts and promoted the emergence of resistant microorganisms, compromising long-term sustainability [6]. Alternatively, plant breeding offers a promising, yet medium- to long-term, solution, whereas the use of elicitors or systemic resistance inducers has emerged as a short-term strategy that harnesses plants’ intrinsic adaptive mechanisms to bolster immunity against biotic stresses [7,8,9].

Plant immunity operates through two primary layers: pattern-triggered immunity (PTI), activated by the recognition of pathogen-associated molecular patterns (PAMPs), and effector-triggered immunity (ETI), resulting from the detection of specific effector proteins secreted by pathogens [8]. Both layers converge to elicit a broad spectrum of defense responses, ranging from physical barriers to biochemical and molecular alterations [10], often orchestrated by complex hormonal signaling networks. Within these networks, salicylic acid (SA) and jasmonic acid (JA) act as central regulators, coordinating distinct but interconnected defense strategies [11,12].

SA plays a central role in defense against biotrophic and hemibiotrophic pathogens, which rely on living host tissues to complete their life cycles [13]. Activation of the SA-dependent signaling pathway induces Systemic Acquired Resistance (SAR), a durable defense state characterized by the accumulation of a subset of pathogenesis-related (PR) proteins, whose induction profiles vary depending on the pathogen, along with enhanced reactive oxygen species (ROS) production, and cell wall reinforcement [12,14]. Key enzymes involved in SAR include β-1,3-glucanase (PR-2), peroxidase (POX), and phenylalanine ammonia-lyase (PAL), which contribute to the synthesis of phenolic compounds and tissue lignification, thereby strengthening structural barriers against pathogen colonization [12,15,16].

In contrast, JA, which may interact synergistically or antagonistically with ethylene (ET) depending on the context, regulates defense against necrotrophic pathogens and herbivorous insects that exploit host cell death [17,18]. The JA/ET-dependent signaling pathway underlies Induced Systemic Resistance (ISR), involving transcriptional activation of defense-related genes, enhanced activity of oxidative enzymes such as lipoxygenase (LOX), and accumulation of secondary metabolites with antimicrobial properties [19,20].

The crosstalk between SA- and JA-mediated pathways is complex and antagonistic, reflecting the need for fine-tuned regulation of immune responses according to pathogen lifestyle and environmental conditions [21,22]. At the molecular level, the integration and fine-tuning of these signaling networks rely on regulatory hubs, including NPR1 (Nonexpressor of Pathogenesis-Related Genes 1), the Mediator complex, and Nudix hydrolases. NPR1 is an important transcriptional co-regulator that integrates SA signaling, turns on PR genes, and ensures that SAR is established.

Its redox-dependent conformational changes allow nuclear translocation and interaction with TGA transcription factors, linking hormonal signaling to gene expression [23,24]. The Mediator complex, a conserved multiprotein assembly, bridges transcription factors and RNA polymerase II, coordinating the transcriptional reprogramming necessary for immune activation and hormonal crosstalk [25,26]. Similarly, Nudix hydrolases (NUDXs) function as metabolic regulators and signaling modulators, maintaining cellular redox homeostasis and contributing to immune priming by balancing ROS production and detoxification [14,27]. Together, these elements form an essential regulatory network that integrates SA- and JA-dependent responses, ensuring specificity and balance in plant immunity [14,21,24,25].

Within this regulatory framework, natural elicitors have garnered attention for their capacity to modulate defense pathways and enhance plant immunity. Humic substances (HS) are recognized as biostimulants that promote both plant growth and resilience to biotic and abiotic stresses [28,29,30]. Recent studies demonstrate that HS can activate both the SA-dependent SAR pathway and the JA/ET-mediated ISR pathway [31,32].

HS represents the most stable and bioactive fraction of natural organic matter, arising from the decomposition and transformation of plant and microbial residues [33]. The chemical composition is heterogeneous and strongly dependent on origin, vermicompost (Vc) or peat (Pt), for example, resulting in variable proportions of aromatic, aliphatic, and oxygenated functional groups [34]. This chemical diversity is crucial for determining the intensity and specificity of HS physiological effects [35,36]. Molecular heterogeneity underlies a wide range of metabolic effects, including enhanced membrane permeability, activation of ion transporters, and stimulation of key processes such as photosynthesis, the Krebs cycle, and ATP and amino acid synthesis [37,38,39,40,41].

Furthermore, HS stimulates enzymatic activities associated with plant defense, such as peroxidase (POX), phenylalanine ammonia-lyase (PAL), and pathogenesis-related protein 2 (PR-2) [32,42], while also modulating reactive oxygen species (ROS) metabolism.

ROS act as signaling molecules that regulate metabolic activities through transduction mechanisms [43]. At the molecular level, HS also influences gene expression in hormonal pathways, functioning as signaling molecules that interact with membrane receptors and trigger canonical defense responses [31,32,39,42,44,45].

Despite these advances, significant gaps remain regarding how HS affect specific plant defense pathways. In particular, the activation of the SA pathway, crucial for defense against biotrophic pathogens such as *Oidium* sp. remains poorly understood. Considering that the HS chemical structure can guide biological interactions, elucidating how substances from different sources influence key regulatory components of the SA pathway, such as *NPR1*, PR genes (*PR-1*, *PR-2*, *PR-4* and *PR-5*), as well as antioxidant enzymes associated with defense signaling. Such clarification is necessary to understand how HS modulate plant immunity. This study aimed to investigate how HS extracted from vermicompost (Vc) and peat (Pt) regulate the SA pathway and defense responses in plants infected with *Oidium* sp. (*On*).

## 2. Results

### 2.1. Characteristics of the Humic Substances Used

Solid-state ^13^C NMR analyses revealed marked differences between HS obtained from Vc and Pt (Figure 1). Vc (Figure 1a) showed a larger area from signals originating from the presence of carbohydrate moieties and peptide linkages centered at 71.7 ppm (O-alkyl C and C–N/O groups). The major signal at 54.3 ppm is often attributed to lignin fragments, as methoxy groups are found in this region. It was possible to observe the presence of unsubstituted aromatic carbons (117.9–126.4 ppm) and carboxylic groups (172.7 ppm). This chemical feature is typical of an organic matter source with a larger proportion of oxygenated compounds, associated with carbohydrate and protein residues. In contrast, the Pt spectrum (Figure 1b) showed a predominance of hydrophobic and aliphatic carbons with an intense signal at 30.5 ppm, along with signals at 120.9 ppm (aromatic carbons) and 170.1 ppm (carboxyl/carbonyl groups). This pattern indicates a more hydrophobic, recalcitrant structure typical of sedimentary organic matter.

The ^13^C CPMAS NMR spectra of HS isolated from Pt and Vc revealed the usual composition of complex natural organic materials with a carbon distribution extended along all the chemical shift range (Figure 1). Vc was characterized by the predominance of CH_3_O/C-N interval and O-alkyl-C components, which represented the 16.7% and 28.1% of the total area, respectively (Table 1), while humic substances isolated from Pt showed the alkyl-C (45–0 ppm) interval corresponding to 50.2% of total spectrum area.

### 2.2. Enzymatic Activity

Considering the chemical differences observed between the HS from Vc and Pt, we investigated whether these contrasts would translate into distinct plant responses. The activity of LOX, PAL, POX, and PR-2, key enzymes involved in plant defense pathways, was evaluated. Application of the HS significantly affected the activity of these defense-related enzymes (Figure 2). LOX activity (Figure 2a) increased in both Vc and Pt treatments compared with the control, with no significant difference between them. However, in the presence of powdery mildew, an increase was observed only in Vc + *On*. Infection strongly induced POX activity (Figure 2b) and further enhanced it in the combined treatments, particularly in Pt + *On*, which exhibited the highest values. PAL activity (Figure 2c) was also stimulated by Vc and Pt, with a more pronounced response in the pathogen-inoculated treatments, where Vc + *On* and Pt + *On* showed higher activity than the control. Finally, PR-2 (Figure 2d) was significantly induced by infection, reaching the highest levels in Pt + *On*, followed by Vc + *On*. Overall, the results indicate that HS, particularly when combined with powdery mildew infection, modulates plant enzymatic activity.

### 2.3. Gene Expression

In general, HS modulates the expression of genes related to systemic defense. In this study, we evaluated 20 genes involved in systemic defense pathways and grouped the results by biological function: co-regulators, hormonal signaling and the phenylpropanoid pathway, antioxidants, and pathogenesis-related (PR) genes.

The treatments differentially modulated the expression of co-regulator genes (Figure 3). *MED25* expression was significantly upregulated in plants treated with HS derived from Pt, whereas pathogen inoculation (*On*) or the combined treatment Pt + *On* led to a pronounced downregulation compared with the control. In contrast, plants treated with HS from Vc or with the combined treatment Vc + *On* maintained expression levels comparable to the control. For *MED16* (Figure 3b), all treatments showed downregulation compared to the control. In *MED14* (Figure 3c), Pt + *On* showed high expression, followed by Vc + *On* and *On*. Plants exposed only to HS from Vc and HS from Pt showed downregulation compared to the controls. Finally, *NPR1* (Figure 3d), a key regulator of systemic acquired resistance, was upregulated in infected plants, with greater upregulation in Vc + *On* and Pt + *On*. Treatments with HS from Vc or HS from Pt alone did not significantly alter transcript regulation compared to the control.

The expression of genes involved in hormonal signaling pathways showed distinct modulation across treatments (Figure 4). In the JA pathway, *LOX1.1* (Figure 4a) and *MYC2* (Figure 4b) were strongly upregulated in plants treated with Vc, with relative expression levels approximately six and fivefold higher than the control, respectively. Both genes also showed moderate induction in Pt + *On* and Vc + *On*, while *On* and Pt alone resulted in lower expression levels. The gene *jar1* (Figure 5c) exhibited significant induction only in Vc, with reduced expression in the remaining treatments. In contrast, *JAZ* (Figure 4d) was highly induced in On, reaching the highest relative expression among treatments, followed by Pt + *On*. Regarding the ET pathway, *ERF* (Figure 4e) showed increased expression in Vc and Pt + *On*, whereas *On* caused a marked reduction compared to the control.

In the SA pathway, the *ICS* and *PAL* genes (Figure 4f,g), both involved in SA biosynthesis and, for PAL, also participating in the phenylpropanoid pathway, displayed contrasting expression profiles. Expression of *ICS* was strongly repressed in On, Vc + *On*, and Pt + *On* treatments compared with the control. In contrast, *PAL* expressions were significantly induced, particularly in the Pt + *On* and Vc + *On* treatments. The *NUDX8* gene (Figure 4h), associated with nucleotide metabolism and SAR, also exhibited a marked upregulation in Pt + *On* and Vc + *On*.

The expression of antioxidant metabolism genes varied among treatments (Figure 5). The APX gene (Figure 5a) showed increased expression in all treatments compared to the control, with the highest transcription levels observed in Pt + *On*, followed by Vc + *On*, Pt, Vc, and *On*. In POX (Figure 5b), expression in Vc + *On* was approximately sevenfold higher than the control. On also promoted upregulation. Meanwhile, Vc and Pt alone showed lower expression levels, similar to the control. The CAT gene (Figure 5c) showed moderate variation between treatments, with a slight induction in Pt and Vc, while *On*, Vc + *On*, and Pt + *On* did not differ significantly from the control. Furthermore, SOD (Figure 5d) was significantly upregulated in Pt + *On*, followed by Vc + *On*, while treatments with *On*, Vc, and Pt resulted in low or similar levels to the control.

The expression of *pathogenesis-related (PR)* genes varied markedly among treatments (Figure 6). The *PR-1* gene (a) showed slight induction in Vc, whereas all other treatments resulted in reduced expression compared with the control, with the lowest transcript levels observed in On and Pt + *On*. The *PR-2* gene (b) was strongly upregulated in Pt + *On* and Vc + *On*, reaching approximately 8- and 6-fold higher levels than in the control, respectively. In contrast, *On*, Vc, and Pt alone exhibited lower expression levels. Similarly, *PR-4* (c) displayed a pronounced induction of Vc + On, followed by Pt + On, whereas the *On*, Vc, and Pt treatments showed only modest increases compared with the control. For *PR-5* (d), expression was highest in Vc, whereas On, Pt, Vc + *On*, and Pt + *On* resulted in intense repression relative to the control.

### 2.4. Severity of the Disease

Phenotypic evaluation confirmed that the structural and functional differences in HS were directly reflected in the plant–pathogen interaction (Figure 7). Seven days after inoculation with *Oidium* sp., control plants showed visible symptoms and a severity index of approximately 12%, while the HS treatments from Vc and Pt showed severity index of 2.5% and 0.8%, respectively. These results complement the results obtained in the enzymatic and molecular analyses.

## 3. Discussion

The results obtained in this study indicate that HS extracted from Vc and Pt, despite exhibiting marked chemical differences, act as multifunctional elicitors of defense responses in tomato (*Solanum lycopersicum* cv. Micro-Tom) infected with *Oidium* sp., a biotrophic pathogen whose interaction with the host depends on the balance between activation and suppression of defense pathways. Both HS stimulated resistance mechanisms; however, the HS from Pt was more effective, reducing disease severity to 0.8%, compared to 2.5% for Vc and 12% in the control (Figure 7).

The tomato cultivar Micro-Tom was selected as a model system due to its high genetic uniformity, short life cycle, compact growth habit, and well-characterized hormonal and defense signaling pathways, which make it widely used in studies of plant immunity and hormone-regulated responses. Importantly, Micro-Tom is considered moderately susceptible to powdery mildew, allowing reliable discrimination of induced resistance responses without masking effects of strong genetic resistance [46,47,48]. This intermediate sensitivity makes it an appropriate biological model to assess the capacity of humic substances to activate defense mechanisms against biotrophic pathogens.

From a chemical standpoint, the greater efficacy of the Pt-derived HS may be associated with its higher hydrophobic character, as indicated by the ~30 ppm peak in the ^13^C NMR spectrum, which is typical of long-chain aliphatic carbons (Figure 1). Hydrophobic domains promote interactions with membranes and may enhance the transport and delivery of signaling molecules to responsive sites, thereby increasing hormonal and enzymatic activation. In contrast, the Vc-derived HS exhibits a higher content of oxygenated polar groups, greater solubility, and a lower degree of hydrophobicity, which may explain its comparatively weaker effects. These findings are consistent with previous studies that have correlated humic hydrophobicity with plant bioactivity [35,36,49]. Spectroscopic and multivariate statistical analyses have demonstrated that fractions with a higher proportion of aliphatic and aromatic hydrophobic structures show stronger correlations with the stimulation of physiological and enzymatic processes in plants [50,51].

The structural basis for the differential bioactivity between peat- and vermicompost-derived humic substances was further supported by solid-state ^13^C CPMAS NMR analyses. The spectra revealed a typical heterogeneous distribution of carbon functional groups, but with marked contrasts between the two materials. Humic substances isolated from peat showed a strong predominance of alkyl-C structures (0–45 ppm), representing 50.2% of the total spectral area, whereas vermicompost-derived HS were enriched in O-alkyl-C (28.1%) and CH_3_O/C–N regions (16.7%) (Table 1). These data indicate that peat HS are enriched in long-chain aliphatic domains, while vermicompost HS contain a higher proportion of carbohydrate-like and oxygenated structures [52,53,54].

The dominant alkyl-C region in peat HS, with intense resonances around 30–33 ppm, is typically associated with long-chain methylene groups derived from waxes, cutin, fatty acids, and other lipid-like structures. These hydrophobic domains are known to enhance molecular interactions with biological membranes and to facilitate the delivery of signaling molecules, thereby increasing the efficiency of defense activation. In contrast, vermicompost HS exhibited stronger signals in the 60–110 ppm region, characteristic of O-alkyl-C groups derived mainly from polysaccharides, hemicellulose, and low-molecular-weight carbohydrates, which confer a more polar and hydrophilic character [54].

These spectral differences were corroborated by structural indices calculated from NMR data, including higher hydrophobicity (HB) and alkyl/O-alkyl (A/OA) ratios in peat-derived HS, reflecting the predominance of apolar aliphatic structures. These chemical features are consistent with the greater biological activity observed for peat HS in this study, as hydrophobic fractions of humic substances have been repeatedly associated with stronger stimulation of enzymatic, hormonal, and defense-related plant responses. Conversely, the lower hydrophobicity and higher carbohydrate-like content of vermicompost-derived HS likely contribute to their reduced, although still significant, capacity to induce resistance responses. Together, the NMR-derived structural differences provide a mechanistic explanation for the distinct biological performance of the two HS sources, linking molecular composition, hydrophobicity, and defense-eliciting capacity in tomato plants.

SAR is a key pathway in defense against biotrophic pathogens [13], mediated by SA. In plants, SA is synthesized via two main pathways: the ICS and PAL pathways. Generally, under pathogen attack, the PAL pathway is preferentially activated, being associated with phenylpropanoid metabolism and the production of phenolics and lignin precursors [55,56,57]. Schiavon and colleagues [58] were the first to show that HS promotes phenylpropanoid metabolism by inducing PAL enzyme activity and expression, thereby increasing total phenolic content in plants. An upregulation of *PAL* gene expression was also observed in citrus plants two days after HS application under non-infectious conditions [32]. In the present study, *ICS* expression remained similar to the control in HS-treated plants and was repressed in the presence of the pathogen. In contrast, *PAL* was strongly induced, with approximately 25× increases in Pt + *On* and 13× in Vc + *On* compared to the control, indicating a metabolic shift favoring the PAL–SA route and enhancing both structural and chemical defenses against the biotrophic fungus.

Activation of the SA pathway was reflected in the induction of classical defense effector genes, particularly those encoding pathogenesis-related (PR) proteins. The expression of *PR-2* and *PR-4* showed marked upregulation, with approximately 6- and 8-fold increases above the control in the combined treatments Pt + *On* and Vc + *On*, respectively, whereas PR-4 expression reached 8- and 5-fold increases in the same treatments. PR-2 (β-1,3-glucanase) and PR-4 (chitinase) enzymes, act synergistically to degrade fungal cell walls and inhibit hyphal growth, mechanisms that are critical for resistance against *Oidium* sp. [59,60]. Conversely, no significant change was observed in *PR-5* expression, a thaumatin-like protein, suggesting that the HS-induced induction was selective for PRs with direct antifungal enzymatic activity rather than those involved in osmotic adjustment or membrane permeability modulation, as typically associated with PR5, such as xylanase inhibition [61]. This expression pattern indicates that HS, particularly those derived from Pt, direct the defense response toward pathways with greater antifungal efficiency, reinforcing their role as selective elicitors of SAR. Supporting this interpretation, previous studies have shown that other SAR-related PRs, including *PR-3*, *PR-7*, and *PR-11*, are also upregulated by HS in citrus plants [32], suggesting that HS-induced transcriptional modulation follows a conserved pattern across plant species and may involve the activation of NPR1-dependent regulatory cofactors, with NPR1 acting as a key positive regulator required for SAR activation. NPR1 is a positive regulator required for the activation of SAR against a wide range of pathogens [23]. NPR1 responds to the redox state and activates SA-dependent genes [24,62].

Nudix proteins also act as regulators in plant immunity. They perform several functions to detect and modulate the levels of their substrates, such as nucleotide sugars, deoxyribonucleoside triphosphates (dNTPs), and mRNAs, in order to maintain proper cellular processes and physiological homeostasis. *NUDX8* acts as another positive regulator of SAR, is associated with SA signaling, and influences redox homeostasis [27,63]. Both *NPR1* and *NUDX8* were positively regulated in plants treated with HS from Pt and Vc, and under the combined treatments from Pt + *On* and Vc + *On*, their expression was amplified. Together, these results support the hypothesis that HS establishes an oxidative priming state that increases the sensitivity and speed of the defense response.

Fine regulation between hormonal pathways was evidenced by the modulation of Mediator complex subunits: *MED14*, *MED16*, and *MED25* exhibited distinct expression patterns. *MED14* (a co-regulator of the SA pathway) was more highly expressed in Pt + *On*, whereas *MED25* (a co-regulator of JA/ET-mediated immunity) was induced by Vc-derived HS but repressed in the combined pathogen treatment. These profiles indicate that Pt-derived HS redirects signaling toward the SA pathway (resistance to biotrophic pathogens), while Vc-derived HS allows a partial balance between SA and JA, a modulation that may explain the intermediate effectiveness observed for Vc. This multiprotein Mediator complex has been recognized as a key regulator of transcriptional reprogramming, functioning as an adapter/co-regulator between sequence-specific transcription factors and RNA polymerase II (RNAPII). The plant Mediator complex is a macromolecular unit composed of 34 subunits, among which MED14, MED16, and MED25 play critical roles in defense pathways against biotic stresses [25,26].

It is also important to highlight the regulation of the JA pathway in this study, which was also investigated due to its antagonistic relationship with the SA pathway and its typical association with defense against necrotrophic pathogens [64,65]. The genes *LOX1*, *MYC2*, and *jar1* (components of JA biosynthesis and signal transduction) were mainly expressed in plants treated with HS from Pt and Vc. The effect of HS on the activation of the JA biosynthetic pathway has already been reported [31,66]. However, in this study, it was demonstrated that under *Oidium* sp. infection, the pathway is repressed. *JAZ*, a repressor protein of JA signaling, showed a substantial increase (~7× in Vc + *On* and ~4× in Pt + *On*). This pattern indicates that, in the presence of a biotrophic pathogen, HS favors the prioritization of the SA pathway through repression of JA effector signaling (via increased *JAZ* expression and decreased *LOX1*/*MYC2*/*jar1*), which is consistent with the classical SA–JA antagonism [67,68] and with the need for plants to suppress responses typically directed against necrotrophs when facing a biotrophic infection. The differentiation between Pt and Vc also suggests that the structural composition of the HS influences the intensity of this prioritization, as Pt induced a stronger suppression of JA signaling than Vc.

Transcriptional activation was accompanied by promising enzymatic and redox responses: significant increases in the activities of PAL, POX, LOX, and PR2 were observed, along with elevations in the expression and activities of the antioxidant system (*APX*, *POX*, *CAT*, *SOD*), particularly in the Pt + *On* and Vc + *On* treatments. The controlled production of reactive oxygen species (ROS) functions as a secondary messenger in defense signaling [69] and contributes to lignin deposition and cell wall crosslinking [70]. Simultaneously, the activation of APX, CAT, and POX is essential to limit oxidative damage and finely modulate the signal. HS from Vc and Pt increased the expression of *APX* and *POX* genes, with even higher expression observed when plants were infected (Pt + *On* and Vc + *On*). Thus, HS influences an integrated response: enhancement of phenolic metabolism (PAL → SA), activation of enzymatic effectors (PRs), and adjustment of the antioxidant system, resulting in both physical and biochemical reinforcement against infection.

Phenotypically, this molecular and biochemical coordination translated into a marked reduction in disease severity. Considering the biotrophic nature of *Oidium* sp., which relies on haustorium formation and maintenance of living tissues, the early activation of oxidative mechanisms, phenol/lignin deposition, and PR protein expression prevents mycelial establishment and the formation of feeding structures, explaining the observed protective effect. Beyond expanding mechanistic understanding, this study represents, to our knowledge, the first report of using HS to control foliar fungal diseases, complementing preexisting evidence of pathogen suppression by HS in soil and field assays [29].

In summary, the HS investigated act as natural inducers of systemic resistance by integrating chemical signals (composition and hydrophobicity), modulating SA biosynthetic pathways (with preference for the PAL pathway), recruiting transcriptional and redox regulators (*NPR1*, *NUDX8*, *MEDs*), and adjusting the balance between SA and JA (reducing *LOX1*/*MYC2*/*jar1* and increasing *JAZ*), together with the activation of the antioxidant system (*APX*, *POX*, *CAT*, *SOD*). Pt-derived HS, due to its higher hydrophobicity, enhances this network of responses, whereas Vc-derived HS induces a more balanced modulation between pathways. These findings position HS as a promising and sustainable tool for integrated management of fungal diseases, with practical implications for the development of bioelicitor formulations and crop protection strategies.

## 4. Materials and Methods

### 4.1. Extraction and Characterization of Humic Substances

Humic substances (HS) were obtained from two sources: (i) vermicompost (Vc), produced from cattle manure using the earthworm *Eisenia andrei*, and (ii) a commercial peat-based product (Pt) supplied by DNAgro Biotechnology. Extraction of HS from Vc was performed with a 5% KOH solution (Sigma-Aldrich, St. Louis, MO, USA) (1:20, *v*/*v*) under agitation for 4 h, followed by centrifugation (15 min, 5000× *g*). The supernatant was neutralized to pH 7.0. Aliquots of each product (Vc and Pt) were frozen and lyophilized for subsequent chemical characterization.

Total organic carbon (TOC) content: performed by dry combustion using a TOC analyzer from Shimadzu (Tokyo, Japan), based on the Dumas combustion method. In this system, each liquid sample is injected into a high-temperature combustion furnace (~680 °C) under an oxygen-rich atmosphere, promoting complete oxidation of all carbon-containing compounds to carbon dioxide (CO_2_). The resulting gases pass through purification traps for moisture and nitrogen oxides, and purified CO_2_ is subsequently quantified by a non-dispersive infrared detector (NDIR). Carbon concentrations were calculated from calibration curves prepared with potassium hydrogen phthalate standards. For TOC measurements, crude humic substance (HS) extracts from vermicompost and peat were diluted 1:1000 using ultrapure water. A final volume of 15 mL was used per reading, with six analytical replicates for each HS source. Ultrapure water, the same used for sample dilution, was included as the analytical blank. Carbon concentration is expressed as mg C L^−1^. The HS extracted from vermicompost contained 1388 ± 12.5 mg C L^−1^ and the peat-derived HS 11,681 ± 151.5 mg C L^−1^, expressed as mean ± standard deviation.

Solid-state ^13^C NMR spectroscopy: spectra were obtained by CP/MAS on a Bruker-400 spectrometer operating at 75.4 MHz for ^13^C. Acquisition parameters included: spectral width of 50 kHz, acquisition time of 50 ms, pulse width of 5 µs (90°), contact time of 1 ms, 4 s recycle delay, ^1H decoupling in gated mode, and 10,000 transients. Hexamethylbenzene (δ = 17.3 ppm) was used as a secondary reference. Signal integrals were calculated in the following chemical shift regions: aliphatic C (0–50 ppm), O-alkyl/peptide C (50–110 ppm), aromatic/olefinic C (110–150 ppm), heteroatom-substituted aromatic C (150–165 ppm), and carbonyl/amidic/ester C (165–200 ppm) [71].

### 4.2. Plant Material and Growth Conditions

Seeds of tomato (*Solanum lycopersicum* cv. Micro-Tom) were obtained from the Laboratory of Hormonal Control and Development, Luiz de Queiroz College of Agriculture, University of São Paulo (ESALQ-USP, Piracicaba, São Paulo, Brazil). The cultivar Micro-Tom is widely recognized as a model system for the study of hormonal signaling pathways in plants. Seeds were surface-sterilized in a 5% NaClO solution containing 2 drops of detergent for 10 min, then rinsed thoroughly with sterile distilled water. The seeds were sown in 300 mL pots containing a 1:1 (*v*/*v*) mixture of commercial substrate and vermiculite, supplemented with 8 g of NPK (10–10–10) per liter of substrate. Plants were grown in a greenhouse under an average temperature of 25 °C and a relative humidity of approximately 60%. The experimental design was completely randomized, with 12 replicates per treatment. Six treatments were established: (i) Control (no HS application and no *Oidium* sp. inoculation); (ii) Pt (peat-derived HS, non-inoculated); (iii) Vc (vermicompost-derived HS, non-inoculated); (iv) Control + *Oidium* sp. inoculation; (v) Pt + *Oidium* sp. inoculation; and (vi) Vc + *Oidium* sp. inoculation.

All HS suspensions were applied at a concentration of 8 mM C L^−1^. The first application was carried out via substrate drenching at the two-true-leaf stage. At 21 days after germination, a second application was performed by foliar spraying. Inoculation with *Oidium* sp. was conducted 48 h after the foliar application of HS. The conidial suspension was obtained by gently scraping infected tomato leaves, and the concentration was adjusted to 1.8 × 10^6^ conidia mL^−1^ using a Neubauer chamber. Plants were uniformly sprayed with the conidial suspension and maintained in the greenhouse under the same environmental conditions described above.

### 4.3. Enzyme Assays

Phenylalanine ammonia-lyase (PAL, EC 4.3.1.5): PAL activity was determined as described by Pascholati et al. [72] with minor modifications. Leaf tissue (100 mg) was ground in liquid nitrogen and homogenized in 10 mL of 0.1 M sodium borate buffer (pH 8.8), supplemented with 5% (*w*/*v*) polyvinylpyrrolidone (PVP) and 1.2 mL L^−1^ β-mercaptoethanol. After centrifugation (10,000× *g*, 25 min, 4 °C), the supernatant was used for the enzymatic reaction, consisting of 1 mL of enzyme extract, 1 mL of 0.1 M phenylalanine, and 1 mL of 0.2 M borate buffer (pH 8.8). Samples were incubated at 36 °C for 60 min in the dark, and the reaction was stopped with 200 µL of 6 M HCl. Absorbance was measured at 290 nm, and activity was expressed as µmol of trans-cinnamic acid min^−1^ g^−1^ fresh weight, using a molar extinction coefficient of ε_290_ = 10^4^.

Peroxidase (POX, EC 1.11.1.7): POX was extracted from leaf tissue ground in liquid nitrogen. Approximately 1 g of tissue was homogenized in 5 mL of 0.1 M sodium acetate buffer (pH 5.0), containing 1% (*v*/*v*) PVP and 1 mM EDTA. The homogenate was centrifuged (10,000× *g*, 10 min, 4 °C), and the supernatant was used for activity determination. The assay was carried out at 30 °C in a reaction mixture containing 2.0 mL of 0.2 M phosphate buffer (pH 5.8), 0.5 mL of 0.38 M H_2_O_2_, 50 µL of 0.02 M guaiacol, and 50 µL of enzyme extract. The increase in absorbance was monitored at 470 nm for 1 min, and activity was expressed as µmol of H_2_O_2_ decomposed min^−1^ g^−1^ fresh weight, using ε_470_ = 26.6 mM^−1^ cm^−1^ [73].

Lipoxygenase (LOX, EC 1.13.11.12): LOX activity was determined according to Axelrod et al. (1981) [74]. Fresh leaves were ground in liquid nitrogen to a fine powder, and the crude extract was used for the assays. The 10 mM sodium linoleate stock solution was prepared from linoleic acid, Tween 20, and 0.5 N NaOH, and stored at −20 °C in light-protected tubes. The reaction mixture contained 1.0 mL of enzyme extract, 4.0 mL of sodium linoleate solution, and 1.0 mL of 50 mM phosphate buffer (pH 6.0). Product formation was monitored spectrophotometrically at 234 nm at 30 s intervals for 120 s. All analyses were performed in triplicate, and enzyme activity was expressed as the change in absorbance min^−1^ g^−1^ fresh weight [74].

β-1,3-glucanase (GLU, EC 3.2.1.29): activity was determined according to Lever (1972) [75] by measuring glucose released from laminarin. Reaction mixtures contained 150 µL of enzyme extract and 150 µL of 2 mg mL^−1^ laminarin solution, incubated at 40 °C for 60 min. Then, 50 µL of the mixture was added to 1.5 mL of p-hydroxybenzoic acid hydrazide (PAHBAH) reagent and heated at 100 °C for 10 min. After cooling to 30 °C in an ice bath, absorbance was measured at 410 nm. Values were corrected against non-incubated controls and quantified using a glucose standard curve. Enzymatic activity was expressed as ng of glucose released min^−1^ mg^−1^ protein [75].

### 4.4. Gene Expression Analysis by RT-qPCR

Total RNA was extracted from leaves of three independent biological replicates using the TRIzol reagent (Invitrogen). RNA quality and quantity were assessed using a NanoDrop 1000 spectrophotometer and by electrophoresis on a 1.5% agarose gel. Complementary DNA (cDNA) was synthesized from 1 µg of total RNA using the GoScript™ Reverse Transcriptase Kit (Promega). Quantitative real-time PCR (RT-qPCR) reactions were performed on a StepOne Fast Real-Time PCR System (Applied Biosystems) using the SYBR Green PCR Master Mix. The cycling conditions were as follows: 95 °C for 2 min, followed by 40 cycles of 95 °C for 20 s and 58 °C for 30 s. Three biological and three technical replicates were analyzed for each gene. Gene expression levels were normalized using actin (Slactin-4) as the reference gene, and relative transcript abundance was calculated using the 2^−ΔΔCT^ method [76]. Primer sequences used for all target genes are provided in Appendix A.

### 4.5. Disease Severity Assessment

Powdery mildew severity in tomato plants was evaluated using the standard area diagram scale proposed by Lage et al. [77], developed for whole tomato leaves exhibiting symptoms of *Oidium* sp. The scale includes six severity levels (1, 5, 10, 20, 40, and 60% of infected leaf area). For each plant, the third and fourth fully expanded leaves below the apex were evaluated, and the percentage of diseased area was visually estimated by direct comparison with the diagrammatic scale. The mean severity per plant was used as the individual severity index (S_p_), and the disease index (DI) for each treatment was calculated as the average of all S_p_ values within that treatment group.

### 4.6. Statistical Analysis

Data on enzymatic activity were subjected to analysis of variance (ANOVA), and when treatment effects were significant (*p* ≤ 0.05), means were compared using the LSD test. Enzymatic analysis statistics were performed in R (R Core Team, version 4.3) using the ExpDes.pt package. Disease severity data, expressed as the percentage of infected leaf area, were analyzed using GraphPad Prism (version 8.1). Although homoscedasticity was not violated (Brown–Forsythe test, *p* = 0.0676), normality assumptions were not met (D’Agostino–Pearson, Shapiro–Wilk, and Kolmogorov–Smirnov tests; *p* < 0.05). Therefore, disease severity was analyzed using the nonparametric Kruskal–Wallis test (*p* < 0.05), followed by Dunn’s multiple-comparison test with Bonferroni adjusted *p* values. Gene expression data were analyzed in GraphPad Prism using one-way ANOVA, and significant differences relative to the control (*p* < 0.05) were indicated as follows: * for induction (higher expression) and ° for repression (lower expression). All ANOVA summary tables are provided in the Appendix A. Graphs were generated using GraphPad Prism.

## 5. Conclusions

In this study, HS demonstrated a multifunctional role in inducing plant defense in tomato plants infected by *Oidium* sp., acting as modulators of hormonal signaling, redox balance, and the expression of defense-related genes. Despite the structural differences observed between HS isolated from Vc and Pt, both sources were equally effective in activating pathways associated with SAR, enhancing the expression of antioxidant genes and PR proteins, and significantly reducing the severity of infection caused by *Oidium* sp., a biotrophic pathogen whose containment strongly depends on mechanisms mediated by SA and SAR. These results reinforce the role of HS as sustainable, integrative biostimulants that coordinate multiple defense pathways and strengthen the plant immune system. Therefore, the use of HS represents a promising strategy to contain the development of Oidium sp. Infection in tomato crops, supporting more sustainable and resilient disease management practices in commercial production systems.

## Figures and Tables

**Figure 1 plants-14-03854-f001:**
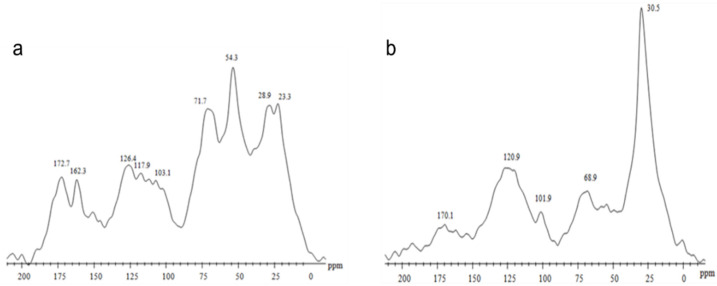
Spectrum of CP/MAS 13C NMR of humic substances (HS) extracted from two sources: (**a**) vermicompost-derived HS obtained from an experimental field area using bovine manure and plant residues, and (**b**) peat-derived HS supplied as a raw commercial humic extract by DNAgro Biotechnology.

**Figure 2 plants-14-03854-f002:**
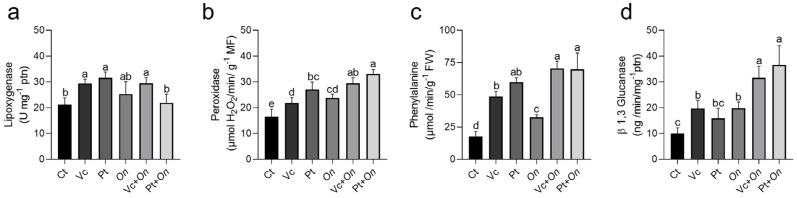
Activity of defense-related enzymes in tomato plants subjected to different treatments. (**a**) LOX, (**b**) POX, (**c**) PAL, and (**d**) PR-2 activities were measured in leaves collected 72 h after treatment and inoculation with *Oidium* sp. Bars represent mean ± standard deviation (SD). Different letters indicate significant differences among treatments by Fisher’s LSD test (*p* < 0.05).

**Figure 3 plants-14-03854-f003:**
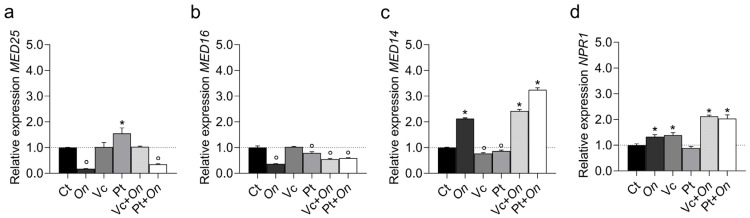
Relative expression of co-regulator genes MED25 (Mediator complex subunit 25) (**a**), MED16 (Mediator complex subunit 16) (**b**), MED14 (Mediator complex subunit 14) (**c**), and NPR1 (Nonexpressor of Pathogenesis-Related genes 1) (**d**) in leaves of tomato cv. Micro-Tom under different treatments: control (Ct), inoculation with Oidium (On), application of humic substances from vermicompost (Vc), humic substances from peat (Pt), vermicompost humic substances and Oidium sp. (Vc + On), and peat humic substances and Oidium sp. (Pt + On). Bars represent mean ± standard deviation (n = 3 biological replicates). Statistical differences in relation to the control were assessed using Dunnett’s multiple comparisons test (*p* < 0.05). An asterisk (*) indicates a significant induction of gene expression relative to the control, while a filled circle (°) indicates a significant repression relative to the control.

**Figure 4 plants-14-03854-f004:**
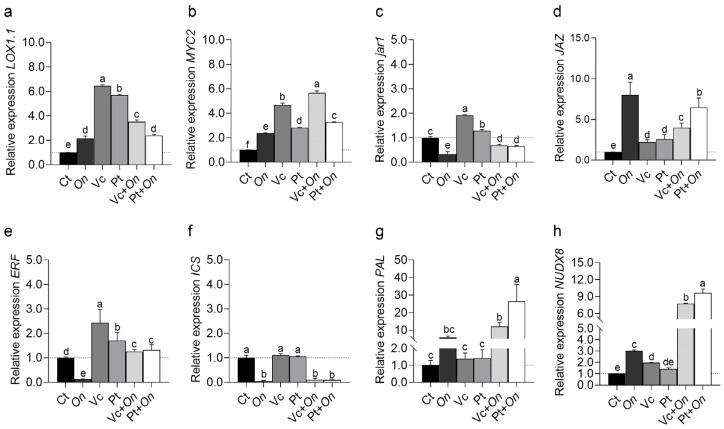
Relative expression of genes involved in JA and SA signaling pathways, ethylene response, and nucleotide metabolism in tomato plants (cv. Micro-Tom) under different treatments. (**a**) *LOX1.1*, (**b**) *MYC2*, (**c**) *JAR1*, and (**d**) *JAZ* are key components of the JA signaling pathway; (**e**) *ERF* is associated with ethylene response; (**f**) *ICS* and (**g**) *PAL* are related to SA biosynthesis; and (**h**) *NUDX8* is involved in NADPH metabolism and stress response. Treatments: control (Ct), inoculation with *Oidium* sp. (On), application of humic substances from vermicompost (Vc) or peat (Pt), and combined treatments (Vc + On, Pt + On). Bars represent mean ± standard deviation (n = 3 biological replicates). Different lowercase letters mean statistical differences in relation to the control were assessed using Dunnett’s multiple comparisons test (*p* < 0.05).

**Figure 5 plants-14-03854-f005:**
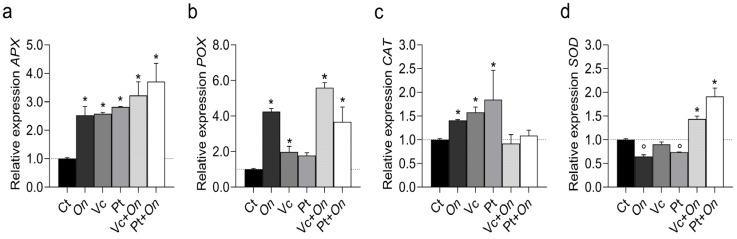
Relative expression of antioxidant-related genes in tomato plants (cv. Micro-Tom) under different treatments. (**a**) *APX* (ascorbate peroxidase), (**b**) *POX* (peroxidase), (**c**) *CAT* (catalase), and (**d**) *SOD* (superoxide dismutase). Treatments: control (Ct), inoculation with *Oidium* sp. (On), application of humic substances from vermicompost (Vc) or peat (Pt), and combined treatments (Vc + *On*, Pt + *On*). Bars represent mean ± standard deviation (n = 3 biological replicates). Statistical differences in relation to the control were assessed using Dunnett’s multiple comparisons test (*p* < 0.05). An asterisk (*) indicates a significant induction of gene expression relative to the control, while a filled circle (°) indicates a significant repression relative to the control.

**Figure 6 plants-14-03854-f006:**
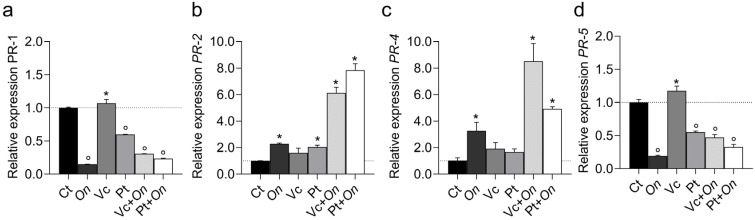
Relative expression of *Pathogenesis-Related* (PR) genes *PR1* (**a**), *PR2* (**b**), *PR4* (**c**), and *PR5* (**d**) in leaves of tomato cv. Micro-Tom under different treatments: control (Ct), inoculation with *Oidium* (*On*), application of humic substances from vermicompost (Vc), humic substances from peat (Pt), vermicompost humic substances and *Oidium* sp. (Vc + *On*), and peat humic substances and Oidium sp. (Pt + *On*). Bars represent mean ± standard deviation (n = 3 biological replicates). Statistical differences in relation to the control were assessed using Dunnett’s multiple comparisons test (*p* < 0.05). An asterisk (*) indicates a significant induction of gene expression relative to the control, while a filled circle (°) indicates a significant repression relative to the control.

**Figure 7 plants-14-03854-f007:**
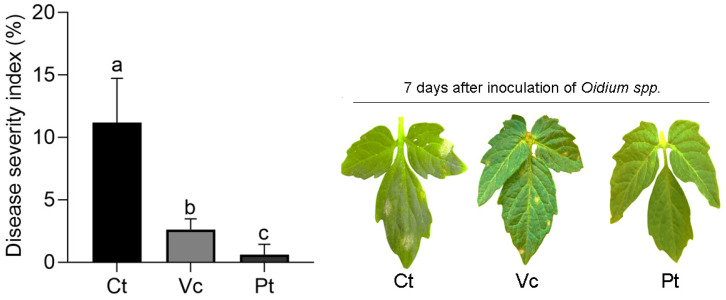
Disease severity index (%) in tomato plants subjected to different treatments (Ct, Vc, and Pt), evaluated 7 days after inoculation with *Oidium* sp. Bars represent mean ± standard error (SE). Different letters above the bars indicate significant differences among treatments according to the Kruskal–Wallis test followed by Dunn’s multiple comparison test with Bonferroni adjustment (*p* < 0.05). Representative leaves from each treatment are shown on the right.

**Table 1 plants-14-03854-t001:** Organic carbon distribution (%) in ^13^C NMR CP/MAS spectra and associated structural indexes.

	C=O	O-aryl-C	aryl-C	O-alkyl-C	CH_3_O/C-N	alkyl-C	HB	A/OA	Ar	LR
	190–160	160–140	140–110	110–60	60–45	45–0				
HS-Pt	10.0	3.5	22.7	9.8	3.7	50.2	3.6	5.1	26.3	1.1
HS-Vc	6.5	8.4	20.3	28.1	16.7	19.8	1.3	0.7	28.7	2.0

The hydrophobic index is the ratio of signal intensities in chemical shift intervals for apolar alkyl and aromatic C components over those of hydrophilic C molecules: (1) HB = Σ[(0–45) + (45–60)/2 + (110–160)]/Σ[(45–60)/2 + (60–110) + (160–190)]. The alkyl ratio determines the relative contribution of apolar versus polar alyl molecules. (2) A/OA = (0–45)/(60–110). The aromaticity index is the total area assigned to aromatic compounds. (3) Ar = Σ[110–160). The lignin ratio relates the area of methoxyl-C + N-alkyl groups to that of O-aryl-C. (4) LR = (45–60)/(145–160).

## Data Availability

The original contributions presented in this study are included in the article/Appendix A. Further inquiries can be directed to the corresponding author.

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
