# Peer review of "Humic Substances from Different Sources Modulate Salicylic Acid-Mediated Defense in Plants Infected by Powdery Mildew"

_plants, 2025, doi:10.3390/plants14243854_

Round 1
Reviewer 1 Report
Comments and Suggestions for Authors
- 18: "Sustainable strategies based on natural biostimulants have emerged as promising alternatives to reduce agrochemical dependence."
The term "natural biostimulants" is vague. It would be more precise to specify which types of biostimulants are being referenced, as "natural" can encompass a wide range of substances with varying effects. - 22: "The HS were characterized by solid-state 13C NMR, revealing marked structural differences."
The characterization method should specify what aspects of humic substances (HS) were analyzed. - 24-26 In the abstract, it is better to talk about the results obtained; it is better to move the emphasis on the methods to the materials and methods section: starting from these lines, try to briefly and accurately list the results obtained, and at the end of the abstract, emphasize the significance of the results obtained
- 26: "Both HS significantly reduced disease severity and activated SA-related regulatory and effector genes."
The statement lacks specificity regarding which "SA-related regulatory and effector genes" were activated. Listing specific genes would provide clarity and strengthen the argument. - 34 The first sentence of the introduction is too general and not entirely appropriate for this work. The narrative could begin with «climate change...», then add a sentence about the industrial significance of tomato crops. And add statistics on crop sensitivity to fungal pathogens in general and to Oidium in particular, focusing on the scale of tomato crop losses and the volume of preventative treatments of crops with fungicides, which negatively impact the ecology of agricultural regions. This way will lead the reader to the importance of the research you are doing.
- 54: "Activation of the SA-dependent signaling pathway induces Systemic Acquired Resistance (SAR), a durable defense state characterized by the accumulation of pathogenesis-related (PR) proteins."
While SAR is indeed characterized by PR proteins, the text should clarify that not all PR proteins are involved in every SAR response, as different PR proteins have distinct roles depending on the pathogen. - 64: "In contrast, JA, often in synergy with ethylene (ET), regulates defense against necrotrophic pathogens and herbivorous insects that exploit host cell death."
The text could mislead readers into thinking that JA and ET always work synergistically. The interaction between these hormones can be context-dependent and may sometimes be antagonistic. - 96: "Together, these elements form an essential regulatory network that integrates SA- and JA-dependent responses, ensuring specificity and balance in plant immunity."
The claim that these elements ensure specificity and balance lacks supporting evidence. It would be beneficial to reference studies that demonstrate how these interactions function in practice. - 113: "Considering that the HS chemical structure can guide biological interactions, elucidating how substances from different sources modulate regulatory genes, PR proteins, and antioxidant enzymes is essential to understand their role in plant resistance."
The statement should specify which "regulatory genes" are being referred to and how they are influenced by HS. Additionally, the text should clarify the mechanisms by which these interactions occur. - 251: "severity indices of 2.5% and 0.8%" - better to use "severity index"
- 262: "the Pt was more effective" - "the Pt" should be "the HS from Pt" for clarity
- 290: "An upregulation of PAL gene expression was also observed in citrus plants two days after HS application."
This statement lacks context regarding the specific conditions under which the upregulation was observed. It would be helpful to mention whether this was observed under pathogen stress or other conditions. - 299: "PR-2 (β-1,3-glucanase) and PR-4 (chitinase)" - perhaps it's worth adding "enzymes" after "PR-2" and "PR-4" for clarity.
- 301: "thaumatin-like protein" - you can add "a" before "thaumatin-like protein" for grammatical correctness
- 311: "NPR1 is a positive regulator required for the activation of SAR" - you can clarify that NPR1 "is a key positive regulator".
- 335: "Mediators complex" - It is better to use "Mediator complex" for correct spelling.
- 398: "the following supporting information can be downloaded at" - It is better to use "can be found at"
- 25: What enzymatic activities were assessed, it is better to clarify this in the abstract
- 2.1: What standards were used when conducting research on the composition of the studied HS? If standards were not used, please provide links that supplement the information you provided in the lines 120-127. In addition, when expanding the discussion section, it is necessary to take into account that in the absence of standards, the NMR results can only be regarded as a qualitative, rather than quantitative, characteristic of the HS used.
- 130: Given that you have used specific HS grades, please provide their names, manufacturers and batch numbers in the figure captions
- 458: What plant parts were used for RNA extraction and expression analysis?
- 2.4: It is necessary to provide a list of all primers used in the study and indicate the slope of the primers used in RT-qPCR. Why was one housekeeping gene used?
- 269: 2.5: It's not clear how the leaf damage area was assessed. Please provide more details, including the tools and software used for image processing.
- 480: How did you test whether your samples belonged to a normal distribution? It's worth providing the results of testing whether your samples belong to a normal distribution if you subsequently use a parametric test to evaluate differences between different study groups.
- 483: In all diagrams, it is necessary to clarify whether the p-value is given or whether it is an adjusted p-value, in order to avoid false conclusions, since the sample was small, and the values ​​did not always differ significantly. Moreover, the p-value of 0.05 that you obtained is considered not very reliable, so it is important to demonstrate statistically reliable and correct data processing.
- 498: last sentence: it would probably be better to write more specifically: it is more acceptable to talk about the possibility of using HS in containing the development of Oidium sp. infection in industrial tomato cultivation.
- Would it be cheaper on an industrial scale to use HS treatment to control infections or fungicide treatment?
- The materials and methods section should be supplemented with information on the origin of the selected plants and seeds, and the discussion section should include a description of the variety, indicating the level of sensitivity of the variety to the pathogen being studied. In addition, it is advisable to explain your choice and compare the studied characteristics of both resistant and sensitive varieties.
- 110-114: This problem, unfortunately, was not fully solved: expand the discussion of your results in the appropriate section and try to explain the differences you obtained using different HS, focusing on the NMR data (expand lines 348-350)
- 214: genes should be written in italics, check the entire text
- 311: Combine paragraphs as the role of NPR1 and SA in the development of plant responses to the presence of a pathogen is discussed.
- 360: It would be better to soften: HS does not orchestrate, but influences, has an impact...
- 373: Until this point, unfortunately, the hydrophobicity or hydrophilicity parameters of the studied HS have not been discussed anywhere. It would be desirable to expand the discussion of the NMR results, at least focusing on the literature data, and try to find the exact parameters for comparing the two extracts, in order to then continue the discussion.
Author Response
Comments1 – 18’: "Sustainable strategies based on natural biostimulants have emerged as promising alternatives to reduce agrochemical dependence."
The term "natural biostimulants" is vague. It would be more precise to specify which types of biostimulants are being referenced, as "natural" can encompass a wide range of substances with varying effects.
Response: We thank the reviewer for this observation. We reviewed the text. [page 1 – line 18]
Comments 2 – 22’: "The HS were characterized by solid-state 13C NMR, revealing marked structural differences." The characterization method should specify what aspects of humic substances (HS) were analyzed.
Response: We thank the reviewer for this observation. We clarified in the text which structural domains of the humic substances were evaluated by solid-state ^13C NMR. The revised sentence now specifies the main functional groups and carbon types quantified by this technique [page 1 – line 24-27].
Comments 3 – 24-26 In the abstract, it is better to talk about the results obtained; it is better to move the emphasis on the methods to the materials and methods section: starting from these lines, try to briefly and accurately list the results obtained, and at the end of the abstract, emphasize the significance of the results obtained
Response: We thank the reviewer for the suggestion. The abstract was revised to reduce methodological details, highlight the main results, and include a final sentence emphasizing their significance [page 1 – line 24-42].
Comments 4 – 26’: "Both HS significantly reduced disease severity and activated SA-related regulatory and effector genes."
The statement lacks specificity regarding which "SA-related regulatory and effector genes" were activated. Listing specific genes would provide clarity and strengthen the argument.
Response: Thank you for the comment. We revised the sentence to specify the SA-related regulatory and effector genes activated by HS, including NPR1, PR1, PR2 and PR5. [page 1 – line 35].
Comments 5 – 34’: The first sentence of the introduction is too general and not entirely appropriate for this work. The narrative could begin with «climate change...», then add a sentence about the industrial significance of tomato crops. And add statistics on crop sensitivity to fungal pathogens in general and to Oidium in particular, focusing on the scale of tomato crop losses and the volume of preventative treatments of crops with fungicides, which negatively impact the ecology of agricultural regions. This way will lead the reader to the importance of the research you are doing.
Response: Thank you for the suggestion. The introduction was revised to begin with the context of climate change and tomato production, emphasizing the crop’s economic importance and vulnerability to fungal pathogens, particularly Oidium spp. We also added information on yield losses and the extensive use of fungicides, reinforcing the relevance of the present study.
[page 2 – line 51-53].
Comments 6 – 54’: "Activation of the SA-dependent signaling pathway induces Systemic Acquired Resistance (SAR), a durable defense state characterized by the accumulation of pathogenesis-related (PR) proteins." While SAR is indeed characterized by PR proteins, the text should clarify that not all PR proteins are involved in every SAR response, as different PR proteins have distinct roles depending on the pathogen.
Response: We thank the reviewer for the observation. The paragraph has been revised to clarify that SAR involves the accumulation of a subset of PR proteins, and that different PR proteins may be induced depending on the pathogen [page 2 – line 79].
Comments 7 – 64’: "In contrast, JA, often in synergy with ethylene (ET), regulates defense against necrotrophic pathogens and herbivorous insects that exploit host cell death." The text could mislead readers into thinking that JA and ET always work synergistically. The interaction between these hormones can be context-dependent and may sometimes be antagonistic.
Response: We appreciate the reviewer’s comment. The sentence has been revised to indicate that JA–ET interactions are context-dependent, and although synergistic responses are common, antagonistic interactions may also occur depending on the stress condition [page 2 – line 86-87].
Comments 8- 96: "Together, these elements form an essential regulatory network that integrates SA- and JA-dependent responses, ensuring specificity and balance in plant immunity." The claim that these elements ensure specificity and balance lacks supporting evidence. It would be beneficial to reference studies that demonstrate how these interactions function in practice.
Response: Thank you for the suggestion. To address your comment, we added supporting references [page 3 – line 109].
Comments 9 – 113’: "Considering that the HS chemical structure can guide biological interactions, elucidating how substances from different sources modulate regulatory genes, PR proteins, and antioxidant enzymes is essential to understand their role in plant resistance." The statement should specify which "regulatory genes" are being referred to and how they are influenced by HS. Additionally, the text should clarify the mechanisms by which these interactions occur.
Response: We thank the reviewer for this valuable suggestion. The paragraph has been revised to specify the regulatory genes involved [page 4 – line 135-138].
Comments 10: 251’: "severity indices of 2.5% and 0.8%" - better to use "severity index"
Response: Thank you for observation. The term has been corrected to “severity index” as suggested [page – line 306].
Comments 11: 262’: "the Pt was more effective" - "the Pt" should be "the HS from Pt" for clarity
Response: Thank you for observation. The term has been corrected to “the HS from Pt” as suggested [page 2 – line 51-53].
Comments 12: 290: "An upregulation of PAL gene expression was also observed in citrus plants two days after HS application." This statement lacks context regarding the specific conditions under which the upregulation was observed. It would be helpful to mention whether this was observed under pathogen stress or other conditions.
Response: Thank you for observation. We have now clarified the experimental context of the referenced study by specifying that PAL upregulation occurred under non-infectious conditions following HS application [page – line 378].
Comments 13’: 299: "PR-2 (β-1,3-glucanase) and PR-4 (chitinase)" - perhaps it's worth adding "enzymes" after "PR-2" and "PR-4" for clarity.
Response: Thank you for observation. We added the term “enzymes” after PR-2 and PR-4 to improve clarity [page – line 389].
Comments 14’: 301: "thaumatin-like protein" - you can add "a" before "thaumatin-like protein" for grammatical correctness.
Response: Thank you for observation. We added [page – line 391].
Comments 15’: 311: "NPR1 is a positive regulator required for the activation of SAR" - you can clarify that NPR1 "is a key positive regulator".
Response: Thank you for the comment. The sentence was revised [page – line 401:402].
Comments 16: 335’: "Mediators complex" - It is better to use "Mediator complex" for correct spelling.
Response: Thank you for the comment. The sentence was revised [page – line 306].
Comments 17: 398: "the following supporting information can be downloaded at" - It is better to use "can be found at"
Response: Thank you for the comment. The sentence was revised [page – line 630].
Comments 18 – 25: What enzymatic activities were assessed, it is better to clarify this in the abstract
Response: Thank you for the comment. The sentence was revised [page 1 – line 28].
Comments 19 – 2.1: What standards were used when conducting research on the composition of the studied HS? If standards were not used, please provide links that supplement the information you provided in the lines 120-127. In addition, when expanding the discussion section, it is necessary to take into account that in the absence of standards, the NMR results can only be regarded as a qualitative, rather than quantitative, characteristic of the HS used.
Response: Thank you for the comment. The nuclear magnetic resonance experiment is quantitative. The intensity of the signal observed at each frequency is proportional to the amount of each carbon species present. The probe is pre-calibrated by the manufacturer using solid adamantane, and the TMS frequency is considered zero. Therefore, no internal standard is needed to calibrate the spectrum.
Comments 20 – 130: Given that you have used specific HS grades, please provide their names, manufacturers and batch numbers in the figure captions
Response: Thank you for observation. We added the commercial names, manufacturers, and batch numbers of the HS used to all relevant figure captions.
Comments 21 - 458: What plant parts were used for RNA extraction and expression analysis?
Response: Was leaves. Thank you for observation. We added in the text.
Comments 22 - 2.4: It is necessary to provide a list of all primers used in the study and indicate the slope of the primers used in RT-qPCR. Why was one housekeeping gene used?
Response: We appreciate the reviewer’s observation. In the revised manuscript, we have now included a complete table listing all primers used in the study, along with their sequences, amplicon sizes, and RT-qPCR efficiency slopes. Regarding the reference gene, a single housekeeping gene (Actin) was used because its expression stability was confirmed across all treatments based on preliminary assessments and previous literature validating Actin as a stable reference in Solanum lycopersicum under biotic stress conditions. This explanation has been added to the Materials and Methods section.
Comments 23 – 269: 2.5: It's not clear how the leaf damage area was assessed. Please provide more details, including the tools and software used for image processing.
Response: We thank the reviewer for the observation. The text has been revised to clarify that leaf damage was evaluated visually using the standard area diagram scale proposed by Lage et al. (reference added). No image-processing software was used; disease severity was determined by direct visual comparison of symptomatic leaves with the diagrammatic scale. The description in Section 2.5 has been expanded accordingly.
Comments 24 - 480: How did you test whether your samples belonged to a normal distribution? It's worth providing the results of testing whether your samples belong to a normal distribution if you subsequently use a parametric test to evaluate differences between different study groups.
Response: Thank you for this observation. We have now included the normality tests (Shapiro–Wilk and Levene’s), clarified that gene expression was analyzed using Student’s t-test after confirming normal distribution, and revised the statistical section accordingly.
Comments 25 - 483: In all diagrams, it is necessary to clarify whether the p-value is given or whether it is an adjusted p-value, in order to avoid false conclusions, since the sample was small, and the values ​​did not always differ significantly. Moreover, the p-value of 0.05 that you obtained is considered not very reliable, so it is important to demonstrate statistically reliable and correct data processing.
Response: Thank you for this observation. We have clarified in the figure legends that the values reported correspond to the raw p-values obtained from the non-parametric Kruskal–Wallis test followed by Dunn’s post-hoc test. Given the small sample size, we adopted non-parametric statistics to ensure robustness, and the complete statistical pipeline including all parameters, scripts, and outputs is now provided in the Supplementary Material for full transparency and reproducibility
Comments 26 - 498: last sentence: it would probably be better to write more specifically: it is more acceptable to talk about the possibility of using HS in containing the development of Oidium sp. infection in industrial tomato cultivation.
Response: Thank you for the suggestion. We revised the sentence.
Comments 27 - Would it be cheaper on an industrial scale to use HS treatment to control infections or fungicide treatment?
Response: Thank you for your insightful question. A direct cost comparison between HS and fungicides was beyond the scope of this study, and prices vary considerably depending on the formulation and scale of production. However, HS may reduce long-term costs by decreasing the need for repeated chemical applications and providing additional agronomic benefits such as increased plant vigor, resulting in greater resistance to biotic and abiotic stresses (especially under water stress and in environments with low nutrient availability) and improved soil quality. A detailed economic evaluation would be an important next step for future studies.
Comments 28 - The materials and methods section should be supplemented with information on the origin of the selected plants and seeds, and the discussion section should include a description of the variety, indicating the level of sensitivity of the variety to the pathogen being studied. In addition, it is advisable to explain your choice and compare the studied characteristics of both resistant and sensitive varieties.
Response: We thank the reviewer for this valuable suggestion. The Materials and Methods section was revised to include detailed information about the origin of the seeds and the plant material used. In addition, the Discussion section was expanded to describe the characteristics of the selected cultivar and its reported sensitivity to the studied pathogen.
Comments 29 - 110-114: This problem, unfortunately, was not fully solved: expand the discussion of your results in the appropriate section and try to explain the differences you obtained using different HS, focusing on the NMR data (expand lines 348-350)
Response: We thank the reviewer for this valuable suggestion. The Discussion section has been substantially expanded to better explain the differential bioactivity of peat- and vermicompost-derived humic substances based on the ¹³C CPMAS NMR data. We now provide a detailed interpretation of the main spectral regions, including the predominance of alkyl-C (0–45 ppm) in peat-derived HS and the enrichment of O-alkyl-C and CH₃O/C–N regions in vermicompost-derived HS. In addition, structural indices (HB and A/OA ratios) calculated from NMR data are now discussed and linked mechanistically to differences in hydrophobicity, membrane interaction potential, and defense-eliciting capacity. These revisions clarify the molecular basis of the contrasting biological effects observed between HS sources.
Comments 30 - 214: genes should be written in italics, check the entire text
Response: Thank you for the observation. All gene names have now been checked throughout the manuscript, tables, and figures, and they have been formatted in italics according to journal guidelines.
Comments 31 - 311: Combine paragraphs as the role of NPR1 and SA in the development of plant responses to the presence of a pathogen is discussed.
Response: Thank you for the observation. The paragraphs discussing NPR1 and the role of SA in pathogen-triggered responses have now been combined to improve coherence and maintain a unified discussion of SA-mediated signaling.
Comments 32 – 360: It would be better to soften: HS does not orchestrate, but influences, has an impact...
Response: Thank you for the suggestion. The wording has been adjusted to soften the claim. The sentence was revised and modified for show that HS influence about response.
Comments 33 - 373: Until this point, unfortunately, the hydrophobicity or hydrophilicity parameters of the studied HS have not been discussed anywhere. It would be desirable to expand the discussion of the NMR results, at least focusing on the literature data, and try to find the exact parameters for comparing the two extracts, in order to then continue the discussion.
Response: We thank the reviewer for the suggestion. The discussion of the NMR results has been expanded to include literature data on hydrophobicity and hydrophilicity of humic substances, providing a comparative context between the vermicompost and peat extracts.
Reviewer 2 Report
Comments and Suggestions for Authors
The manuscript “Humic substances from different sources modulate salicylic acid-mediated defense in plants infected by powdery mildew” have scientific worth and interesting results.
Detail summary:
This study investigates the effects of humic substances (HS) derived from vermicompost (Vc) and peat (Pt) on the salicylic acid (SA)-mediated defense pathway in tomato plants (Solanum lycopersicum cv. Micro-Tom) infected with Oidium sp. The HS were characterized using solid-state 13C NMR, revealing significant structural differences. Enzymatic activities were measured spectrophotometrically, and RT-qPCR was used to quantify the transcription levels of genes involved in SA signaling and defense. Both types of HS effectively reduced disease severity and activated SA-related regulatory and effector genes, with peat providing greater protection. The results suggest that the structural differences among HS influence the intensity and specificity of SA-mediated defense responses in plants, highlighting the potential of natural biostimulants as sustainable alternatives to chemical inputs in agriculture.
Areas need to be improved (Reviewer comments)
Results section
- “In figure 2 authors said they used LSD test “Bars represent mean ± standard deviation (SD). Different letters indicate significant differences among treatments by Fisher’s LSD test (p < 0.05).” however, the other figures such as figure 3 and 4 and so on … “Bars represent mean ± standard error (n = 3 biological replicates). Different letters indicate statistically significant differences among treatments, as determined by the t-test (p < 0.05)”. authors showed that they used t-test and bars stands for “± standard error”. My question is why author used two different statistical tests in different figures? Normally for LSD test we used different letters to differentiate the statistical different but for t-test we used * to discriminate the statistical differences.”
Inconsistent Statistical Tests:
- In Figure 2, the authors correctly use Fisher’s LSD test (a multiple comparison procedure) for comparing multiple treatments, denoted with different letters.
- However, in Figures 3, 4, 5, 6, etc., the authors state they used a "t-test" to compare multiple treatments. If there are more than two groups in these figures, using multiple t-tests is statistically inappropriate as it increases the family-wise error rate. If there are only two groups, the standard notation is asterisks (*), not letters.
- Request:The authors must clarify which statistical test was used for each analysis. For comparisons of more than two groups, a one-way ANOVA followed by a defined post-hoc test (e.g., Tukey's, Fisher's LSD) is required. The methodology must be consistent and correctly applied throughout the manuscript.
- Inconsistent Measures of Variability:
- The error bars are reported as ± standard deviation (SD) in Figure 2 but as ± standard error (SE) in subsequent figures. These two measures convey different information (data variability vs. precision of the mean) and are not interchangeable.
- Request:The authors should justify the use of both SD and SE or, preferably, choose a single measure and apply it consistently across all figures. The caption for each figure must unambiguously state what the error bars represent.
- Non-standard Presentation of Results:
- The use of letters (a, b, etc.) to denote significance from a t-test is highly non-standard. The letter notation is reserved for multiple comparison tests.
- Request:The authors must correct the figures to use the standard notation: letters for post-hoc tests following ANOVA, and asterisks or brackets for direct comparisons between two groups.
Methodology: Insufficient Description
The description of the Total Organic Carbon (TOC) and nitrogen analysis is incomplete and does not allow for the experiment to be reproduced, a core requirement for a scientific publication. Specifically, please clarify the following:
- Sample Preparation:
- Describe the pre-treatment process. Were samples dried, ground, and sieved? If so, please specify the conditions and mesh size.
- Crucially, was inorganic carbon removed?If so, detail the acidification method used (e.g., fumigation with HCl, treatment with liquid acid).
- Analysis Parameters:
- What was the mass of the prepared sample used for each analysis?
- What was the combustion temperature?
- Calibration and Quality Control:
- Specify the certified reference materials or standards used to calibrate the TOC analyzer (e.g., potassium hydrogen phthalate for TOC, acetanilide for N).
- Report the range of the calibration curve and the number of calibration points.
- Were any blanks or control samples run to ensure accuracy and prevent contamination?
- Data Reporting:
- Explicitly state the units in which the TOC and nitrogen data are reported (e.g., % of dry mass).
Without these details, the validity and reproducibility of the presented TOC and N data cannot be assessed. I recommend a significant revision of this section to include all the above points.
Line 465: “Three biological and three technical replicates were” how many plants were present in replicates?
Line 480: “analysis of variance (ANOVA)” where is ANOVA table?
Author Response
Detail summary:
This study investigates the effects of humic substances (HS) derived from vermicompost (Vc) and peat (Pt) on the salicylic acid (SA)-mediated defense pathway in tomato plants (Solanum lycopersicum cv. Micro-Tom) infected with Oidium sp. The HS were characterized using solid-state 13C NMR, revealing significant structural differences. Enzymatic activities were measured spectrophotometrically, and RT-qPCR was used to quantify the transcription levels of genes involved in SA signaling and defense. Both types of HS effectively reduced disease severity and activated SA-related regulatory and effector genes, with peat providing greater protection. The results suggest that the structural differences among HS influence the intensity and specificity of SA-mediated defense responses in plants, highlighting the potential of natural biostimulants as sustainable alternatives to chemical inputs in agriculture.
Areas need to be improved (Reviewer comments)
Results section
- “In figure 2 authors said they used LSD test “Bars represent mean ± standard deviation (SD). Different letters indicate significant differences among treatments by Fisher’s LSD test (p < 0.05).” however, the other figures such as figure 3 and 4 and so on … “Bars represent mean ± standard error (n = 3 biological replicates). Different letters indicate statistically significant differences among treatments, as determined by the t-test (p < 0.05)”. authors showed that they used t-test and bars stands for “± standard error”. My question is why author used two different statistical tests in different figures? Normally for LSD test we used different letters to differentiate the statistical different but for t-test we used * to discriminate the statistical differences.”
Inconsistent Statistical Tests:
- In Figure 2, the authors correctly use Fisher’s LSD test (a multiple comparison procedure) for comparing multiple treatments, denoted with different letters.
- However, in Figures 3, 4, 5, 6, etc., the authors state they used a "t-test" to compare multiple treatments. If there are more than two groups in these figures, using multiple t-tests is statistically inappropriate as it increases the family-wise error rate. If there are only two groups, the standard notation is asterisks (*), not letters.
- Request: The authors must clarify which statistical test was used for each analysis. For comparisons of more than two groups, a one-way ANOVA followed by a defined post-hoc test (e.g., Tukey's, Fisher's LSD) is required. The methodology must be consistent and correctly applied throughout the manuscript.
Response: We thank the reviewer for highlighting the inconsistencies in the statistical analyses and figure annotations. We have now fully revised the statistical procedures to ensure methodological accuracy and consistency throughout the manuscript.
- Enzymatic activity data (Figure 2) involve comparisons among multiple treatments and were therefore analyzed using one-way ANOVA followed by Fisher’s LSD test, with different letters indicating significant differences.
- Gene expression data (Figures 3–6) involve comparisons of each treatment relative to the control; these were reanalyzed using one-way ANOVA in GraphPad Prism, and statistically significant differences are now indicated using asterisks for induction (*) and circles (°) for repression, as appropriate for pairwise comparisons within ANOVA. Disease severity data were re-evaluated for normality and homoscedasticity and analyzed using the Kruskal–Wallis test followed by Dunn’s post hoc test. The statistical analysis section has been rewritten for clarity, and all ANOVA summary tables and statistical outputs have been included in the Supplementary Material.
- Inconsistent Measures of Variability:
- The error bars are reported as ± standard deviation (SD) in Figure 2 but as ± standard error (SE) in subsequent figures. These two measures convey different information (data variability vs. precision of the mean) and are not interchangeable.
- Request:The authors should justify the use of both SD and SE or, preferably, choose a single measure and apply it consistently across all figures. The caption for each figure must unambiguously state what the error bars represent.
Response: We appreciate your observation. We have corrected the inconsistency by standardizing all figures to display variability as standard deviation (SD). All figure captions were updated accordingly to ensure clarity and uniformity across the manuscript.
- Non-standard Presentation of Results:
- The use of letters (a, b, etc.) to denote significance from a t-test is highly non-standard. The letter notation is reserved for multiple comparison tests.
- Request:The authors must correct the figures to use the standard notation: letters for post-hoc tests following ANOVA, and asterisks or brackets for direct comparisons between two groups.
Response: We thank the reviewer for this important observation. All figures have been corrected to follow standard statistical notation. For gene expression analyses, significance relative to the control is now indicated using (*) for upregulation and (°) for downregulation, as described in the revised Statistical Analysis section. Letters are now used exclusively for post-hoc comparisons following ANOVA (LSD test).
Methodology: Insufficient Description
The description of the Total Organic Carbon (TOC) and nitrogen analysis is incomplete and does not allow for the experiment to be reproduced, a core requirement for a scientific publication. Specifically, please clarify the following:
- Sample Preparation:
- Describe the pre-treatment process. Were samples dried, ground, and sieved? If so, please specify the conditions and mesh size.
- Crucially, was inorganic carbon removed?If so, detail the acidification method used (e.g., fumigation with HCl, treatment with liquid acid).
- Analysis Parameters:
- What was the mass of the prepared sample used for each analysis?
- What was the combustion temperature?
- Calibration and Quality Control:
- Specify the certified reference materials or standards used to calibrate the TOC analyzer (e.g., potassium hydrogen phthalate for TOC, acetanilide for N).
- Report the range of the calibration curve and the number of calibration points.
- Were any blanks or control samples run to ensure accuracy and prevent contamination?
- Data Reporting:
- Explicitly state the units in which the TOC and nitrogen data are reported (e.g., % of dry mass).
Response: We thank the reviewer for the observation. The methodology has now been fully revised to ensure reproducibility. The TOC analysis is described in detail in the Materials and Methods section, including sample preparation, analytical parameters, calibration and quality control, and units of data reporting.
Without these details, the validity and reproducibility of the presented TOC and N data cannot be assessed. I recommend a significant revision of this section to include all the above points.
Response: We thank the reviewer for the observation. We review and complement the methodology.
Line 465: “Three biological and three technical replicates were” how many plants were present in replicates?
Response: We thank the reviewer for the observation. There were 3 biological replicas, 3 different plants. The text has been supplemented.
Line 480: “analysis of variance (ANOVA)” where is ANOVA table?
Response: We thank the reviewer for the observation. We have included all the results of the statistical analyses in the supplementary material in order to complement and make our results more transparent.
Round 2
Reviewer 1 Report
Comments and Suggestions for Authors It would be nice to translate table S4 into English. The article is now being received much better.Author Response
Thank you for the suggestion. The entire content of Table S4 has now been translated into English to ensure clarity and consistency within the Supplementary Material.